# Consensus Report of the Technical-Scientific Associations of Italian Dental Hygienists and the Academy of Advanced Technologies in Oral Hygiene Sciences on the Non-Surgical Treatment of Peri-Implant Disease

**DOI:** 10.3390/ijerph20032268

**Published:** 2023-01-27

**Authors:** Alessio A. Amodeo, Andrea Butera, Marco Lattari, Giulia Stablum, Antonia Abbinante, Maria Teresa Agneta, Jacopo Lanzetti, Domenico Tomassi, Stefania Piscicelli, Maurizio Luperini, Arcangela Colavito, Lorella Chiavistelli, Rita Politangeli, Matteo Castaldi, Gianna Maria Nardi

**Affiliations:** 1RDH DHA, Department of Biomedical, Surgical and Dental Sciences, University of Milan, 20122 Milan, Italy; 2RDH DHA, IRCCS Foundation, Ca’Granda General Hospital in Milan, 20122 Milan, Italy; 3Unit of Dental Hygiene, Section of Dentistry, Department of Clinical, Surgical, Diagnostic and Paediatric Sciences, University of Pavia, 27100 Pavia, Italy; 4RDH, Free Lancer in Pavia, 27110 Pavia, Italy; 5RDH, AIDI (Associazione Igienisti Dentali Italiani), University of Bari, 70121 Bari, Italy; 6RDH DHA, AIDI, 70121 Bari, Italy; 7RDH, University of Turin, 10124 Torino; 8RDH DHA, Catholic University of the Sacred Hear, 00168 Roma, Italy; 9RDH DHA, UNID, University of Modena and Reggio Emilia, 41121 Modena, Italy; 10RDH DHA, ATASIO (Accademia Tecnologie Avanzate nelle Scienze di Igiene Orale), 70121 Bari, Italy; 11RDH DHA, Department of Oral and Maxillofacial Sciences, Sapienza University of Rome, 00161 Rome, Italy

**Keywords:** consensus report, peri-implant mucositis, peri-implantitis, non-surgical perimplant therapy, adjunctive therapy, air polishing, laser, photodynamic therapy, ozone

## Abstract

Background: The recent publication of the new classification of periodontal and peri-implant disease has given clear indications on the parameters to be taken into consideration to correctly diagnose the different phases of these diseases. To date, however, there are no equally clear indications on the treatments to be implemented to solve these diseases. The objective of this Consensus Report is to provide guidance for the non-surgical management of peri-implant mucositis and peri-implantitis. For the drafting of the consensus, the most recent scientific literature was analysed. Materials and Methods: A group of 15 expert Italian dental hygienists were selected by the Italian technical-scientific societies (AIDI, UNID and ATASIO) and, starting from the literature review, they formulated indications according to the GRADE method (Grading of Recommendations, Assessment, Development, and Evaluation, a tool for rating the quality of evidence, used to draw up systematic reviews and clinical guidelines) on the treatment of peri-implant mucositis, peri-implantitis and on management of the various implanting surfaces. Conclusions: in accordance with the international literature, non-surgical therapy alone can resolve peri-implant mucositis, but not peri-implantitis. Several adjunctive therapies have been considered and some appear to be helpful in managing inflammation.

## 1. Introduction

A new classification for peri-implant health, peri-implant mucositis and peri-implantitis was developed by the World Workshop on the Classification of Periodontal and Peri-implant Diseases and Conditions in 2017. During the workshop, clear clinical criteria were identified to define peri-implant health, peri-implant diseases, and relevant aspects of implant site conditions and deformities. The aim of the workshop was to reach a consensus for this classification that could be accepted worldwide. As for periodontal health, peri-implant health is characterized by an absence of visual signs of inflammation and bleeding on probing [1].

The use of dental implants for supporting prosthetic rehabilitations has shown highly satisfactory results regarding restoration of the patient’s function and aesthetics, as well as in terms of long-term survival. However, dental implants can lose supportive bone caused by local inflammation during peri-implant diseases [2].

Peri-implant mucositis is defined by bleeding on probing and visual signs of inflammation, there is strong evidence that peri-implant mucositis is caused by plaque, and it can be reversed with measures aimed at eliminating the biofilm [3]. Peri-implantitis is marked as a plaque-associated pathologic condition occurring in the tissue around dental implants, characterized by inflammation in the peri-implant mucosa and subsequent progressive loss of supporting bone [4].

The management and non-surgical treatment of peri-implant disease is an issue that still divides the scientific community. Therefore, the prevention and treatment of peri-implant diseases are important aspects in clinical dentistry, and the available scientific evidence should help define adequate preventive and therapeutic approaches. The limited available literature suggests that mechanical non-surgical therapy could be effective in the treatment of peri-implant mucositis [5].

The primary objective for treatment of both peri-implant mucositis and peri-implantitis is elimination of biofilm from the implant surface, however it can be challenging. Biofilm formation is partially controlled by an interbacterial communication mechanism that is dependent on bacterial population density, called quorum sensing.

By doing so, efficient mechanical debridement is difficult but critical in the management of dental implant infections. The prosthetic supra-structure often prevents effective cleaning around the implant neck by the patient, and conventional mechanical therapies adopted from the treatment of periodontal disease have their limitations because getting good access to the relevant area can be difficult [6].

As guidelines have never been published, clinical practice may apply techniques that deviate from current scientific evidence.

This consensus aims to highlight the importance and need for scientific evidence in clinical decision-making in the treatment of patients with peri-implant mucositis and peri-implantitis.

Its main objective is therefore to support daily clinical practice with evidence-based recommendations for the various interventions used in the different phases of non-surgical therapy, based on the best available evidence and/or expert consensus.

## 2. Materials and Methods

During the A.T.A.S.I.O. (Academy of Advanced Technologies in Oral Hygiene Sciences, note of the translator) National Congress on 5–6 February 2021, the participating dental hygienists were asked to take an instant poll with the aim of surveying their attitudes towards the diagnosis and non-surgical management of peri-implant problems in their daily practice. The analysis of the answers showed that the non-surgical management of mucositis and peri-implantitis is largely in line with the attitude that the current scientific literature proposes, but there are still some grey areas on the choice of certain instruments and techniques in addition to the standard instrumentation. For this reason, it was decided, based on a thorough literature review, to propose a series of clinical recommendations to guide clinicians in the daily management of peri-implant disease therapy.

### 2.1. Focused Questions

The aim of this consensus is to highlight the importance and need for scientific evidence in clinical decision-making in the treatment of patients with peri-implant mucositis and peri-implantitis.

### 2.2. Eligibility Criteria

Type of studies. Randomized controlled clinical trials, prospective clinical trials, and Meta-analysis were included.

Types of participants. Participants with the peri-implant disease were considered.

Type of interventions. Evaluation of the scientific literature about the efficacy of different therapy and techniques applied in patient affected by peri-implant disease.

All publications that did not meet the eligibility criteria, all studies published not in English, all studies for which the full text was not available were excluded.

### 2.3. Search Strategy

A panel of 12 expert dental Hygienists, representative of both the university area and the free profession, was selected by the (Academy of Advanced Technologies in Oral Hygiene Sciences, note of the translator) (ATASIO) and the two Technical Scientific Associations (ATS) listed by the Ministry of Health, the (Association of Italian Dental Hygienists, note of the translator) (AIDI) and the (National Union of Dental Hygienists, note of the translator) (UNID), in order to discuss the clinical results proposed by the current scientific literature (Table 1).

The panellists signed the conflict of interest declaration and then a first plenary meeting was convened on a zoom platform, in which the session chairman explained the objectives, working methodology and criteria for inclusion and exclusion of scientific articles.

The bibliography search was carried out by the authors, no manual search was performed and only publications written in English were searched in three databases: MEDLINE/PubMed, EMBASE and Cochrane.

### 2.4. Research

We performed the search using the following terms: “peri-implant oral health”, “peri implant-disease”, “peri-implant mucositis”, “peri-implantitis”, “non-surgical peri-implant therapy”, “peri-implant Clinical Practice Guidelines”, “peri-implant disease management”, “manual instrumentation”, “manual instrumentation AND peri-implant disease”, “manual instrumentation AND peri-implant mucositis”, “manual instrumentation AND peri-implantitis”, “manual instrumentation AND non-surgical peri-implant therapy” “glycine”, “glycine AND peri-implant diseases”, “glycine AND peri-implant mucositis”, “glycine AND peri-implantitis”, “glycine AND non-surgical peri-implant therapy”, “erythritol”, “erythritol AND peri-implant diseases”, “erythritol AND peri-implant mucositis”, “erythritol AND peri-implantitis”, “erythritol AND non-surgical peri-implant therapy”, “laser”, “laser AND peri-implant diseases”, “laser AND peri-implant mucositis”, “laser AND peri-implantitis”, “laser AND non-surgical peri-implant therapy”, “photodynamic therapy”, “photodynamic AND peri-implant disease”, “photodynamic AND peri-implant mucositis”, “photodynamic AND peri-implantitis”, “photodynamic AND non-surgical peri-implant therapy”, “chlorhexidine”, “chlorhexidine AND peri-implant diseases”, “chlorhexidine AND peri-implant mucositis”, “chlorhexidine AND peri-implantitis”, “chlorhexidine AND non-surgical peri-implant therapy”, “ozone”, “ozone AND peri-implant diseases”, “ozone AND peri-implant mucositis”, “ozone AND peri-implantitis”, “ozone AND non-surgical peri-implant therapy”.

### 2.5. Eligibility and Conflicts of Interests

Two reviewers selected eligible studies by examining the list of titles and abstracts and considering the inclusion and exclusion criteria. Full articles from eligible titles and abstracts were obtained and independently reviewed to determine eligibility. Discrepancies between these reviewers regarding the selection and inclusion of any specific paper were discussed until a consensus was reached or a third reviewer determined inclusion or exclusion.

A second meeting was organized at which the final document was presented for approval. According to the principles provided by the Guidelines International Network (Schunemann et al., 2015), working group members who declared relevant and potential conflicts of interest abstained from voting on the recommendations in this consensus.

### 2.6. Evaluation of the Collected Data

The second step saw the reviewers present the data to the panel members, who voted anonymously to confirm or not that they agreed with the current scientific evidence; the vote was done with a dichotomous option: agree/disagree.

It was decided to proceed with anonymous voting in order not to influence the voting in any way and to make practitioners free to vote against techniques strongly supported by the literature, this to understand whether the clinical attitude reflects the evidence or not. In the case of a lack of evidence, the experts were asked to state their thoughts.

A table containing the strength of the recommendation, the degree of recommendation of the procedure and the anonymous panel vote was compiled.

The levels of available evidence (evidence) and the strength of recommendations were classified according to the National Plan Guidelines:I.evidence based on a meta-analysis of randomised controlled trials.II.evidence based on at least one randomised controlled trial.III.evidence based on at least one non-randomised controlled study.IV.evidence based on at least one non-controlled experimental study.V.evidence based on non-experimental descriptive studies (including comparative studies).VI.evidence based on strong consensus and/or expert clinical experience.

The strength of the recommendations was then classified as follows:A.the performance of that particular diagnostic procedure or test is strongly recommended. This indicates a particular recommendation supported by good quality scientific evidenceè, although not necessarily type I or II.B.there is some doubt as to whether that particular procedure or intervention should always be recommended, but it is felt that its performance should be carefully considered.C.there is substantial uncertainty for or against the recommendation to perform the procedure or intervention.D.performing the procedure is not recommended.E.performing the procedure is strongly discouraged.

### 2.7. Targets

#### 2.7.1. Target Users of the Guideline

Dental and medical professionals, together with all stakeholders related to health care, particularly oral health, including patients.

#### 2.7.2. Targeted Environments

Dental and medical academic/hospital environments, clinics, and practices.

#### 2.7.3. Targeted Patient Population

People with peri-implant mucositis and periimplantitis.

## 3. Results

### 3.1. Peri-Implant Mucositis

Based on the new classification of periodontal and peri-implant disease, the diagnosis of peri-implant mucositis requires the presence of bleeding and/or suppuration at gentle probing with or without increased probing depth compared to previous examinations, and the absence of bone loss beyond crestal bone level changes resulting from initial bone remodelling.

Based on current knowledge and evidence, the expert panel produces the recommendations shown in Table 2.

### 3.2. Perimplantitis

Based on the new classification of periodontal and peri-implant disease, the diagnosis of peri-implantitis requires the presence of bleeding and/or suppuration at probing, increased probing depth compared to baseline and the presence of bone loss in addition to crestal bone resorption resulting from initial bone remodelling.

Based on current knowledge and evidence, the expert panel produces the recommendations shown in Table 3.

### 3.3. Implant Surface Management

The management of surfaces influences the long-term maintenance of dental implants.

A correct choice of the headmasters in terms of materials of the hand tools and implants surfaces is fundamental to solve implant-prosthetic problems without creating iatrogenic damage.

Based on current knowledge and evidence, the expert panel produces the recommendations shown in Table 4.

In order to better understand the recommendations of the panel of experts, the studies taken into consideration are summarized in Table 5 and Table 6.

## 4. Discussion

Dental implants are a valid support in dental restorations and are part of the oral cavity of a significant proportion of the population.

Peri-implant diseases are becoming increasingly prevalent, the prevalence of peri-implant mucositis ranged from 19 to 65% and peri-implantitis ranged from 1 to 47%, according to systematic reviews and meta-analysis conducted in recent years [48].

With the increasing use of dental implants, peri-implant diseases are also becoming more prevalent, therefore the prevention of peri-implant disease is an important aspect to take care of.

In recent years, several protocols for the non-surgical management of mucositis and peri-implantitis have been proposed in the literature, the aim of which is the decontamination of implant surfaces by mechanical debridement as a basis for the reduction of bacterial colonisation and the elimination of the risk factor for peri-implant disease, including adherent oral biofilm.

Over the years, protocols have been proposed involving the addition of low-abrasiveness powders with air polishing systems, the use of laser and photodynamic therapy, the addition of antiseptics (chlorhexidine) both locally and as an additional home therapy, and even the addition of ozone.

The aim of our consensus was to, based on an analysis of the most recent literature, provide indications for the non-surgical treatment of peri-implant pathologies.

The effectiveness of the various types of treatment of peri-implantitis in addition to or as an alternative to mechanical debridement is still debated in the literature.

Regarding peri-implant mucositis, considering the evidence, we can state that non-surgical therapy alone is sufficient to resolve the inflammation. As far as additional therapies are concerned, there is no evidence that the use of other technologies and systems in the literature alone gives better results than conventional therapy.

For the instrumentation of implant sites, the use of dedicated instruments such as titanium curettes and peek inserts for ultrasound is recommended. Steel instruments are not recommended due to their hardness, which can scratch implant surfaces, and Teflon instruments due to the risk of losing some splinters during instrumentation.

Regarding the use of additional therapies to NSPT, long-term antimicrobial effects and reduction of inflammation around implant sites have been shown with the use of low-abrasiveness powders (such as erythritol and glycine), photodynamic therapy, antiseptic substances, and the use of ozone therapy.

Regarding peri-implantitis, evidence suggests that non-surgical therapy alone is often not sufficient to resolve inflammation. As with the treatment of mucositis, the use of titanium and peek instruments is recommended for the management of peri-implantitis, so as not to alter the surfaces of fixtures and abutments.

Current evidence suggests that, in addition to non-surgical therapy, the use of low-abrasiveness powders (such as erythritol and glycine), laser, photodynamic therapy and ozone therapy can be a valuable adjunct to mechanical debridement to control inflammation.

It is also essential to know and recognise the different types of implant surfaces to choose the most appropriate instruments and avoid possible iatrogenic damage.

The use of abrasive pastes or low abrasive powders such as glycine or erythritol does not appear to alter implant surfaces, as do non-metallic hand instruments or sonic/ultrasonic instruments with dedicated tips.

The use of metal instruments, whether manual or sonic, should always be avoided/however, it is not always possible to access all sites due to the size of the instrument or the anatomical surface of the implant that is difficult to decontaminate.

## 5. Conclusions

Based on the results discussed in this consensus report, we can assume that the additional therapies found in scientific literature and used for peri-implantitis disease, may provide additional clinical benefits in the non-surgical treatment of peri-implant diseases.

Analysing the clinical, microbiological and radiographical effects of those therapies, supporting the mechanical debridement for the treatment of peri-implant diseases, some improvements have emerged.

Considering the results found in scientific literature, the application alone of these additional therapies is not recommended, however their application in addition to mechanical debridement with non-metallic hand tools or sonic/ultrasonic instruments is helpful.

## Figures and Tables

**Table 1 ijerph-20-02268-t001:** Panel of 12 expert dental Hygienists.

Organisers/Scientific Associations	Delegates
**Academy of Advanced Technologies in Oral Hygiene Sciences (ATASIO)**	
	Gianna Maria NARDI; President of ATASIO; Associate Professor Department of Oral and Maxillofacial Sciences, Sapienza University of Rome
	Lorella CHIAVISTELLI, ATASIO Board Member
	Rita POLITANGELI, ATASIO Ordinary Member
	Matteo CASTALDI, ATASIO Ordinary Member
	Arcangela COLAVITO, ATASIO Board Member
**Methodologists**	Alessio AMODEO, Andrea BUTERA, Marco LATTARI, Giulia STABLUM
**Technical Scientific Associations (ats)**	
**Association of Italian Dental Hygienists (AIDI)**	
	Antonia ABBINANTE, President of AIDI (Associazione Igienisti Dentali Italiani); Director of professional educational activities Degree Course Dental Hygiene University of Bari;
	Maria Teresa AGNETA, board member of AIDI
	Jacopo LANZETTI, board member of AIDI
**National Union of Dental Hygienists (UNID)**	
	Domenico TOMASSI, Past President of UNID (Unione Nazionale Igiensiti Dentali); Director of professional educational activities Degree Course Dental Hygiene Catholic University of the Sacred Hear
	Stefania PISCICELLI, Vice-President of UNID; Adjunct Professor Degree Course Dental Hygiene Catholic University of the Sacred Hear;
	Maurizio LUPERINI, President of UNID; Adjunct Professor Degree Course Dental Hygiene Univeristy of Modena and Reggio Emilia

**Table 2 ijerph-20-02268-t002:** Peri-Implant Mucositis Recommendations.

*Q1A:*	*Is Causal Therapy Effective in the Non-Surgical Management of Peri-Implant Mucositis?*
*Recommendation based on expert consensus*	Current scientific evidence shows that peri-implant mucositis can be successfully treated with non-surgical therapy. Non-surgical peri-implant treatment usually includes mechanical debridement of oral biofilm and calculus. Scaling and root planing (SRP) peri-implant mucositis sites, using curettes and ultrasound devices in titanium or polyether-ether-ketone (PEEK) coated tips, with or without antimicrobials, has been shown a significant statistical reduction in inflammation index such as Bleeding on Probing (BoP) on peri-implant tissues. Titanium curettes are recommended for debridement of implant surfaces, as carbon fibre and Teflon curettes, due to their brittleness, tend to break easily, sometimes leaving shavings in the peri-implant sulcus. The use of steel curettes is not recommended, as their hardness can alter implant surfaces.
*Literature to Support*	Figuero E et al., 2014 [2], Suárez-López Del Amo F et al., 2016 [7], Menezes M K et al., 2016 [8], Philip et al., 2022 [9], Butera et al., 2022 [10], Dommisch H et al., 2022 [11]
*Degree of recommendation*	**I** (evidence based on a meta-analysis of randomized controlled trials)
*Strength of consensus*	**A** (the performance of that particular diagnostic procedure or test is strongly recommended.
** *Q2A:* **	** *Are the results of causal therapy better after the use of low abrasive powders alone or in addition to standard instrumentation?* **
*Recommendation based on expert consensus*	Current scientific evidence suggests the use of low-abrasiveness powders (glycine and erythritol) in addition to mechanical therapy. The use of air-polishing systems alone does not seem to give better results than conventional therapy. Several scientific studies confirmed no significant improvements of the application of low abrasive powders alone or in addition to NSPT, however a long-term reduction of PPD (Probing Depht), BoP (Bleeding on Probing) and PI (Plaque Index) were evaluated in both treatment groups but not as a replacement of NSPT.
*Literature to Support*	Butera et al., 2022 [10], Schwarz F et al., 2015 [12], Sun F et al., 2022 [13], Daubert M D et al., 2019 [14], Ji Y et al., 2014 [15]
*Degree of recommendation*	**I** (evidence based on a meta-analysis of randomized controlled trials)
*Strength of consensus*	**A** (the performance of that particular diagnostic procedure or test is strongly recommended.
** *Q3A:* **	** *Are the results of treatment with the additional application of the laser superior to non-surgical instrumentation alone?* **
*Recommendation based on expert consensus*	Current clinical studies do not demonstrate a further improvement in results with the addition of laser therapy to conventional therapy; however, there is evidence that the antimicrobial action of laser-assisted technologies significantly reduces inflammation around implant surfaces due to the antimicrobial action performed; therefore, it can be a valuable support before or after conventional therapy. Er:YAG, CO_2_ and Diode lasers with wavelength of 980nm were studied as an adjunction of NSPT, no statistically significant improvements were evaluated regarding of clinical parameters such as PPD and BoP. Current evidence shows laser therapy in combination with surgical/non-surgical therapy provided minimal benefit in PPD reduction, CAL gain and PI reduction in the treatment of peri-implant diseases.
*Literature to Support*	Butera et al., 2022 [10], ChalaM et al., 2020 [16], Lin G et al., 2018 [17], Albaker M A et al., 2018 [18], Tenore G et al., 2020 [19]
*Degree of recommendation*	**II** (evidence based on at least one randomized controlled trial)
*Strength of consensus*	**A** (the performance of that particular diagnostic procedure or test is strongly recommended.)
** *Q4A:* **	** *Are the results of treatment with the additional application of photodynamic therapy superior to non-surgical instrumentation alone?* **
*Recommendation based on expert consensus*	Current clinical studies do not demonstrate a further improvement in results with the addition of photodynamic therapy to conventional therapy; however, there is evidence that the antimicrobial action of photodynamic therapy significantly reduces inflammation around implant surfaces due to the antimicrobial action performed; therefore, it can be a valuable support before or after conventional therapy. Scientific evidence demonstrates a minimal statistically significant reduction in scores of mPI (p < 0.001), mBI (p < 0.001), PD (p < 0.001) in both groups. Limited evidence suggests that PDT may represent a valuable tool in attaining inflammation reduction on a short-term basis in peri-implant diseases but only as a support of NSPT.
*Literature to Support*	ChalaM et al., 2020 [16], Albaker M A et al., 2018 [18], Sculean A et al., 2021 [20], Shetty B et al., 2022 [21], Nardi G.M. et al., 2022 [22]
*Degree of recommendation*	**II** (evidence based on at least one randomized controlled trial)
*Strength of consensus*	**A** (the performance of that particular diagnostic procedure or test is strongly recommended.)
** *Q5A:* **	** *Does the additional use of antiseptics (chlorhexidine) or its locally usage, improve the clinical outcome of non-surgical instrumentation alone?* **
*Recommendation based on expert consensus*	The results of the current scientific evidence indicate that adjunctive therapy with CHX does not improve the clinical results obtained with non-surgical management of peri-implant mucositis alone; due to the properties of selectivity, substantivity and penetrability, it can be considered a valid support before, during and after conventional non-surgical therapy. Current evidence claims no statistically differences intragroup regarding of clinical inflammatory indices such as PPD, GBI, PI and BoP, however a decrease and modulation of oral microbiome were identified with a persistence of aerobic microbiome after 1 months of CHX in addition to NSPT.
*Literature to Support*	Menezes M K et al., 2016 [8], Philip et al., 2022 [9], Butera et al., 2022 [10], Scwarz F. et al., 2015 [12], Liu S et al., 2020 [23], Butera et al., 2022 [24], McKenna F D et al., 2013 [25]
*Degree of recommendation*	**I** (evidence based on a meta-analysis of randomized controlled trials)
*Strength of consensus*	**A** (the performance of that particular diagnostic procedure or test is strongly recommended.
** *Q6A:* **	** *Does the additional use of ozone improve the clinical outcome of non-surgical instrumentation alone?* **
*Recommendation based on expert consensus*	Current scientific evidence shows significant improvements with the use of ozone in addition to mechanical therapy. The use of ozone alone does not seem to give better results than conventional therapy. The administration of gaseous ozone or topical gel ozone applied in the implant pocket seems to reduce Red Complex pathogens and inflammatory indices, such as BoP, but considered as a great potential of his application in addition to NSPT.
*Literature to Support*	Butera et al., 2022 [10], McKenna F D et al., 2013 [25], Wychowański P et al., 2021 [26], Nardi GM et al., 2022 [27]
*Degree of recommendation*	**II** (evidence based on at least one randomized controlled trial)
*Strength of consensus*	**B** (there is some doubt as to whether that particular procedure or intervention should always be recommended, but it is felt that its performance should be carefully considered)

**Table 3 ijerph-20-02268-t003:** Peri-Implantitis recommendations.

*Q1B:*	*Is Causal Therapy Effective in the Non-Surgical Management of Peri-Implantitis?*	
*Recommendation based on expert consensus*	Non-surgical treatment of peri-implantitis provides an improvement in clinical parameters such as BoP (20–50%) and PPD (≤1 mm). However, in advanced cases, complete resolution of the disease is unlikely. Titanium curettes are recommended for debridement of implant surfaces, as carbon fibre and Teflon curettes, due to their brittleness, tend to break easily, sometimes leaving shavings in the peri-implant sulcus. The use of steel curettes is not recommended, as their hardness can alter implant surfaces.	
*Literature to Support*	Figuero E et al., 2014 [2], Dommisch H et al., 2022 [11], Renvert S et al., 2019 [28], Zhao P et al., 2021 [29]	
*Degree of recommendation*	**I** (evidence based on a meta-analysis of randomized controlled trials)	
*Strength of consensus*	**A** (the performance of that particular diagnostic procedure or test is strongly recommended.	
** *Q2B:* **	** *Are the results of causal therapy better after the use of low abrasive powders alone or in addition to standard instrumentation?* **	
*Recommendation based on expert consensus*	Current scientific evidence suggests the use of low-abrasiveness powders (glycine and erythritol) in addition to mechanical therapy. The use of air-polishing alone significantly reduces probing bleeding on peri-implant implants; therefore, it is considered a fundamental support in the non-surgical therapy of peri-implantitis. Clinical evidence claim a statistically improvements in both groups (NSPT and administration of glycine in addition to NSPT) in terms of clinical, radiographical and microbiological aspects.	
*Literature to Support*	Butera et al., 2022 [10], Schwarz F et al., 2015 [12], Sun F et al., 2022 [13], Hentenaar D et al., 2021 [30]	
*Degree of recommendation*	**I** (evidence based on a meta-analysis of randomized controlled trials)	
*Strength of consensus*	**A** (the performance of that particular diagnostic procedure or test is strongly recommended.	
** *Q3B:* **	** *Are the results of treatment with the additional application of the laser superior to non-surgical instrumentation alone?* **	
*Recommendation based on expert consensus*	According to current scientific evidence, it is recommended not to use lasers exclusively for the non-surgical therapy of peri-implantitis. There are beginning to be RCTs in the literature showing benefits from the addition of certain types of lasers (Diode with wavelength 980 nm, CO_2_ and Erbium), the reduction of inflammation prior to non-surgical therapy certainly improves patient and operator comfort during TNC. However, no significant statistically reduction of clinical indices such as PPD and BoP was evaluated even in a short-term period (3 months) for the application of the laser alone. A minimal reduction of PPD, GI and CAL gain was evaluated in both groups demonstrate a great potential of lasers in addition to NSPT.	
*Literature to Support*	Figuero E et al., 2014 [2]; Butera et al., 2022 [10], Lin G et al., 2018 [17], Tenore G et al., 2020 [19], Mattar H et al., 2021 [31]; Scwarz F. et al., 2015 [32]; Wang C et al., 2021 [33], Atieh M A et al., 2022 [34]	
*Degree of recommendation*	**I** (evidence based on a meta-analysis of randomized controlled trials)	
*Strength of consensus*	**A** (the performance of that particular diagnostic procedure or test is strongly recommended.	
** *Q4B:* **	** *Are the results of treatment with the additional application of photodynamic therapy superior to non-surgical instrumentation alone?* **	
*Recommendation based on expert consensus*	Current scientific evidence suggests that the addition of photodynamic therapy to the mechanical treatment of peri-implantitis should be considered, although there are no articles indicating its effective use apart from mechanical therapy. A-PDT was confirmed to be a safe treatment for peri-implantitis, and the short-term efficacy of a-PDT in addition to NSPT was evaluated. However, its efficacy remains restricted since no statistically reduction in term of clinical and microbiological indices such as Red Complex pathogens, PPD, BoP and CAL gain was marked.	
*Literature to Support*	Sculean A et al., 2021 [20], Lin Y et al., 2022 [35], Ohba et al., 2020 [36], Zhao Y. et al., 2021 [37], Romeo U et al., 2016 [38]	
*Degree of recommendation*	**I** (evidence based on a meta-analysis of randomized controlled trials)	
*Strength of consensus*	**B** (there is some doubt as to whether that particular procedure or intervention should always be recommended, but it is felt that its performance should be carefully considered.)	
** *Q5B:* **	** *Does the additional use of antiseptics (chlorhexidine) or his locally usage improve the clinical outcome of non-surgical instrumentation alone?* **	
*Recommendation based on expert consensus*	The results of current scientific evidence indicate that adjunctive therapy with CHX does not improve the clinical results obtained with non-surgical management of peri-implantitis alone. Clinical evidence demonstrated that the adjunctive CHX therapy had no significant effect on BOP reduction, PPD and CAL gain even in a short- or long-term follow-up. The addition of CHX to mechanical debridement, compared with mechanical debridement alone, did not significantly enhance the clinical results.	
*Literature to Support*	Butera et al., 2022 [10], Liu S et al., 2020 [23], Butera et al., 2022 [24], Zhao P et al., 2021 [29], Scwarz F. et al., 2015 [32]	
*Degree of recommendation*	**I** (evidence based on a meta-analysis of randomized controlled trials)	
*Strength of consensus*	**B** (there is some doubt as to whether that particular procedure or intervention should always be recommended, but it is felt that its performance should be carefully considered).	
** *Q6B:* **	** *Does the additional use of ozone improve the clinical outcome of non-surgical instrumentation alone?* **	
*Recommendation based on expert consensus*	Current scientific evidence suggests that the addition of ozone to the mechanical treatment of peri-implantitis should be considered, although there are no articles indicating its effective use apart from mechanical therapy. The administration of gaseous ozone or topical gel ozone applied in the implant pocket seems to reduce Red Complex pathogens and inflammatory indices, such as BoP, but considered as a great potential of his application in addition to NSPT.	
*Literature to Support*	Butera et al., 2022 [10], McKenna F D et al., 2013 [25], Wychowański P et al., 2021 [26]	
*Degree of recommendation*	**II** (evidence based on at least one randomized controlled trial)	
*Strength of consensus*	**B** (there is some doubt as to whether that particular procedure or intervention should always be recommended, but it is felt that its performance should be carefully considered)	

**Table 4 ijerph-20-02268-t004:** Recommendations for the management of implant surface.

*Q1C:*	*Is the Use of Rubber Pads on Smooth and Rough Surfaces Recommended?*	
*Recommendation based on expert consensus*	Handling of smooth surfaces using rubber pads without the use of abrasive pastes does not create iatrogenic damage; however, it is not always possible to access all anatomical areas due to their size. The handling of rough surfaces with the use of grommets is approved, clinical studies show that grommets with abrasive paste do not cause iatrogenic damage to smooth surfaces; however, the procedure is difficult to perform in submucosal areas.	
*Literature to Support*	Louropoulou A et al., 2011 [39]; Renvert S et al., 2017 [40], Fais L et al., 2012 [41], Zul Fahmi Bahari et al., 2021 [42], Al-Hashedi A A et al., 2019 [43], Yen Nee W et al., 2022 [44]	
*Degree of recommendation*	**III** (evidence based on at least one non-randomized controlled study);	
*Strength of consensus*	**B** (there is some doubt as to whether that particular procedure or intervention should always be recommended, but it is felt that its performance should be carefully considered)	
** *Q2C:* **	** *Is the use of non-metallic hand tools recommended for smooth surfaces and rough surfaces?* **	
*Recommendation based on expert consensus*	The use of non-metallic hand tools on smooth surfaces does not cause significant surface damage; however, it is not always possible to access the surfaces to be decontaminated and easily remove hard deposits. The use of non-metallic hand instruments on rough surfaces does not cause damage to the surfaces; however, it is not always possible to access the surfaces to be decontaminated and easily remove hard deposits, the materials currently available may flake off in the submucosa area leaving fragments that could cause inflammation.	
*Literature to Support*	Louropoulou A et al., 2011 [39]; Renvert S et al., 2017 [40], Yen Nee W et al., 2022 [44], Gehrke et al., 2014 [45]	
*Degree of recommendation*	**III** (evidence based on at least one non-randomized controlled study;)	
*Strength of consensus*	**B** (there is some doubt as to whether that particular procedure or intervention should always be recommended, but it is felt that its performance should be carefully considered)	
** *Q3C:* **	** *Is the use of metal hand tools recommended on smooth and rough surfaces?* **	
*Recommendation based on expert consensus*	The use of metal hand instruments is strongly discouraged because of the iatrogenic injuries these inserts cause to smooth surfaces. The use of metallic hand instruments causes iatrogenic injuries on rough surfaces, and there is not always easy access to the submucosal environment.	
*Literature to Support*	Louropoulou A et al., 2011 [39]; Renvert S et al., 2017 [40], Yen Nee W et al., 2022 [44], Gehrke et al., 2014 [45]	
*Degree of recommendation*	**III** (evidence based on at least one non-randomized controlled study);	
*Strength of consensus*	**E** (performing the procedure is strongly discouraged)	
** *Q4C:* **	** *Is the use of low-abrasiveness powders on smooth and rough surfaces recommended?* **	
*Recommendation based on expert consensus*	The use of airpolish with low-abrasiveness powder does not damage surfaces, the airpolish technology allows easy access to surfaces even in anatomically unfavourable situations.	
*Literature to Support*	Louropoulou A et al., 2011 [39]; Renvert S et al., 2017 [40], Yen Nee W et al., 2022 [44], Gehrke et al., 2014 [45], Menini M et al., 2021 [46], Menini M et al., 2019 [47],	
*Degree of recommendation*	**III** (evidence based on at least one non-randomized controlled study);	
*Strength of consensus*	**A** (the performance of that particular diagnostic procedure or test is strongly recommended.	
** *Q5C:* **	** *Is the use of sonic/ultrasonic with a non-metallic tip on smooth and rough surfaces recommended?* **	
*Recommendation based on expert consensus*	The use of non-metallic instruments on sonic/ultrasonic technology can be considered for the removal of soft and hard deposits, access to the submucosal area is not always easy due to the size of the inserts. The use of non-metallic instruments on sonic/ultrasonic technology can be considered for the removal of soft and hard deposits, access to the submucosal area is not always easy due to the size of the inserts.	
*Literature to Support*	Louropoulou A et al., 2011 [39]; Renvert S et al., 2017 [40], Yen Nee W et al., 2022 [44]	
*Degree of recommendation*	**III** (evidence based on at least one non-randomized controlled study);	
*Strength of consensus*	**B** (there is some doubt as to whether that particular procedure or intervention should always be recommended, but it is felt that its performance should be carefully considered)	
** *Q6C:* **	** *Is the use of sonic/ultrasonic with a metal tip on smooth and rough surfaces recommended?* **	
*Recommendation based on expert consensus*	The use of metal instruments on sonic/ultrasonic technology is strongly discouraged due to the iatrogenic injuries these inserts cause to smooth surfaces.	
*Literature to Support*	Louropoulou A et al., 2011 [39]; Renvert S et al., 2017 [40], Yen Nee W et al., 2022 [44], Gehrke et al., 2014 [45]	
*Degree of recommendation*	**III** (evidence based on at least one non-randomized controlled study);	
*Strength of consensus*	**E** (performing the procedure is strongly discouraged)	
** *Q7C:* **	** *Is the use of titanium hand instruments recommended for smooth and rough surfaces?* **	
*Recommendation based on expert consensus*	The use of titanium hand instruments is strongly discouraged because of the iatrogenic injuries these inserts cause to smooth surfaces. The use of titanium hand instruments is not always possible to easily access the submucosal environment.	
*Literature to Support*	Louropoulou A et al., 2011 [39]; Renvert S et al., 2017 [40], Yen Nee W et al., 2022 [44], Gehrke et al., 2014 [45]	
*Degree of recommendation*	**III** (evidence based on at least one non-randomized controlled study);	
*Strength of consensus*	**E** (performing the procedure is strongly discouraged)	

**Table 5 ijerph-20-02268-t005:** Summary of RCT’s Studies.

Investigators	Study Design	Materials & Method	Patients	Parameters Involved	Follow up (Week)
Menezes M K et al., 2016 [8]	Double Blind RCT	the 37 patients were divided into test group (basic periodontal therapy +0.12% chlorhexidine) with 61 implants; and control group (basic periodontal therapy + placebo) with 58 implants.	37 patients	PI, GBI, BOP, PPD	4, 12 and 24 weeks
Philip et al., 2022 [9]	Double Blind RCT	The 89 patients with at least one implant diagnosed with peri-implant mucositis were randomly assigned to one of three study groups: delmopinol (0.2% delmopinol hydrochloride containing decapinol mouthwash), chlorhexidine (containing 0.2% chlorhexidine digluconate) or a placebo.	89 patients	PI, BOP and PPD. and microbiological sample	12 weeks
Sun F et al., 2022 [13]	RCT	The patients in the test group received mechanical submucosal debridement using titanium curettes combined with application of glycine powder air-polishing, while the control group received mechanical submucosal debridement using titanium only.	28 patients with 62 implants	PI, BOP and PPD	8 weeks
Ji Y et al., 2014 [15]	Pilot RCT	All patients, after receiving instructions and motivation, were sorted to SRP. They were then divided into 2 groups, and the test group underwent decontamination of the peri-implant site with glycine air-polishing.	24 patients with 33 implants	PI, BI, PPD	12 weeks
Tenore G et al., 2020 [19]	RCT	The patients were randomly divided into two groups; control group (CG) received conventional non-surgical treatment and test group (TG) received conventional non-surgical treatment and diode laser application.	23 patients	BOP and PPD	12 weeks
Shetty B et al., 2022 [21]	CCT	all patients underwent full-mouth SRP and peri-implant mechanical debridement. subsequently the patients of the test group were treated with photodynamic therapy.	34 patients	PI, BI, PPD and subgingival oral yeasts colonization.	12 weeks
Butera et al., 2022 [24]	RCT	The 20 patients with mucositis were treated with TPNC and subsequently the patients from Group 1 were treated with Curasept Periodontal Gel (chlorhexidine 1%) at the sites of peri-implant mucositis in quadrants Q1 and Q3, while Biorepair Parodontgel Intensive (Lactobacillus Ferment and Lactoferrin) was used for quadrants Q2 and Q4. The quadrants were reversed for patients in Group 2.	20 patients	PI, GBI, BOP, PPD, Marginal Mucosal Conditions (MMC)	24 weeks
McKenna F D et al., 2013 [25]	Double Blind RCT	All implant sites were guarded against toothbrushing by inserting a guard or a gum shield during toothbrushing. In parallel with inducing peri-implant mucositis, computer randomization was used to expose the gingival crevice of each implant site into one of the following treatments: (1) O_3_ and saline (0.9% NaCl), (2) H_2_O_2_ (3%) and air (O_2_), (3) O_3_ and H_2_O_2_, (4) air and saline (control group). All four implant sites in each patient were treated with a different sequence of the four treatments.	20 patients with 80 implants	PI, BI, GI	21 days
], Hentenaar D et al., 2021 [30]	RCT	One group of patients was treated once with an air polisher using erythritol-based powder (grain size 14 μm) containing 0.3% chlorhexidine The other group patients were treated once with the piezoelectric ultrasonic scaler with a Polyether Ether Ketone (PEEK)-coated plastic tip (PI instrument, EMS).	80 patients with 139 implants	IP, BOP, PPD, MBL, microbiological	1 year
Scwarz F. et al., 2015 [32]	Prospective case series	Seventeen patients (24 implants) were diagnosed with peri-implant mucositis and received mechanical debridement + local antiseptic therapy using chlorhexidine digluconate (MD + CXH), while 17 patients (21 implants) diagnosed with peri-implantitis were assigned to Er:YAG laser therapy.	34 patients with 45 implants	PI, BOP and PPD	24 weeks
Ohba et al., 2020 [36],	RCT	Patients with pus discharge from a peri-implant pocket were randomized into two groups, the irrigation and a-PDT groups. The peri-implant pocket was irrigated by normal saline in the irrigation group, and a saline irrigation and subsequent a-PDT was performed in the a-PDT group.	22 patients with 26 implants	PI, BOP, Pus Discharge, K-Mucosa, PPD,	7 days
Fais L et al., 2012 [41]	In Vitro Study	The titanium discs were divided into 6 groups which reproduced some situations like daily oral hygiene manoeuvres. the 6 groups are: IW—Immersion in deionised water EN—Immersion in fluoride-free toothpaste IFT—Immersion in fluoride toothpaste BW—Brushing with deionised water BT—Brushing fluoride-free toothpaste BFT—Brushing fluoride toothpaste	72 disks, 36 Ti and 36 Ti-6Al-4V	surface topography and surface roughness	
Zul Fahmi Bahari et al., 2021 [42]	A Scanning Electron Microscopy Study	The implant implants were mounted in stone supports and randomly divided into 3 groups which were Control (C) (*n* = 5), Airflow (AF) (*n* = 5), and Rubber cup with powder respectively. pumice (RC). All fixtures were subjected to 2 minutes of prophylactic procedures.	15 BEGO Semados^®^ implant		
Al-Hashedi A A et al., 2019 [43]	In vitro study	Thermoplastic co-polyester splints (1 mm thick) covering all maxillary teeth were produced. The splints were used to fix the Ti discs at the buccal aspect of the premolar and molar areas, each splint housed for 12 Ti discs. The participants were asked to wear the splints for 24 h in order to allow for soft biofilm to accumulate on the Ti surfaces. all discs were then decontaminated with 2 different prophylaxis pastes.	24 titanium discs divided into 2 patients,		
Gehrke et al., 2014 [45]	In vitro study	Half of the discs had a machined surface (group 1), while the other half had their surfaces treated with TiO_2_ particles followed by acid etching (group 2). To remove the artificial calculus, four methods were tested: (M1)—scraping with a Teflon curette; (M2)—scraping with a titanium curette; (M3)—cleaning with an air-powder abrasive system; and Method 4 (M4)—cleaning with an ultrasonic scaler with a metal tip	50 titanium discs	surface topography and surface roughness	
Menini M et al., 2021 [46]	RCT	The patients were divided into three groups, each of which received two hygiene therapies randomly administered in each hemiarch using a split-mouth design. The possible treatments were: group 1: glycine powder air polishing and use of sponge floss vs sponge floss only in. group 2: glycine powder air polishing vs use of an ultrasonic device with a PEEK fibre tip coating in; group 3: glycine powder air polishing vs use of carbon fibre curettes and sponge floss in.	85 patients with 357 implants	PI, BOP, PPD	
Menini M et al., 2019 [47]		three different professional oral hygiene treatments were applied: - glycine powder air polishing - sodium bicarbonate powder air polishing - manual scaling with carbon-fiber curettes.	30 patients with 32 implants	PI, SB, BOP, PPD.	

**Table 6 ijerph-20-02268-t006:** Summary of Systematic Review and Meta-Analysis.

Investigators	Problem	Intervention/Comparison	Outcomes
Figuero E et al., 2014 [2]	Describe the different approaches to treat peri-implant diseases.	Only RCTs were selected, and the evaluation was divided into three sections: therapy of peri-implant mucositis; nonsurgical therapy of peri-implantitis; and surgical therapy of peri-implantitis. In addition to mechanical debridement, some additional therapies have also been considered.	Mechanical Debridement is effective in resolving peri-implant mucositis. For peri-implantitis, on the other hand, it is not always sufficient and the use of additional therapies or a surgical approach is suggested.
Suárez-López Del Amo F et al., 2016 [7]	Evaluate the effectiveness of non-surgical therapy for the treatment of peri-implant diseases.	the effectiveness of mechanical debridement and some additional therapies in the resolution of inflammation was evaluated, considering the improvement of some parameters such as PPD, IP, BOP.	Non-surgical treatment for peri-implant mucositis seems to be effective while modest and not-predictable outcomes are expected for peri-implantitis lesions.
Butera et al., 2022 [10]	evaluate adjunctive therapies to mechanical debridement in patients with peri-implant disease.	The experimental group was assisted by one or more laser treatments such as diode lasers, Er: YAG laser, Nd: YAG laser, Er, Cr: YSGG laser, LLLT (Low-Level Laser Therapy), PDT (Photodynamic therapy); ozone treatments such as ozone gas, ozone water, ozone gel; treatments with probiotics such as Lactobacillus or Bifidobacterium; treatments with glycine and erythritol air-polishing or perio-polishing; chlorhexidine treatments such as chlorhexidine mouthwash or gel.	Some therapies such as air-polishing with glycine/erythritol, ozone therapy, chlorhexidine and probiotics seem to be able to give some benefits.
Dommisch H et al., 2022 [11]	To evaluate the efficacy of professionally administered chemical agents as an adjunctive treatment to sub-marginal instrumentation.	The studies were selected by comparing the results in the management of peri-implant inflammation through mechanical therapy alone or with the addition of topical antibiotics, topical antiseptics and aPDT.	it is concluded that the additional application of aPDT, 0.95% NaOCl or 0.12% CHX does not further improve the changes in BOP and/or PD compared to SRP alone.
Schwarz F et al., 2015 [12]	evaluate the efficacy of non-surgical therapy in patients with peri-implant mucositis and peri-implantitis, also evaluating treatments with alternative or additional measures.	the efficacy of some additional therapies, such as air polishing, antiseptics, and antibiotics, in the resolution of inflammation was evaluated, considering the improvement of some parameters such as PPD, IP, BOP.	While mechanical debridement alone has been shown to be effective for the management of peri-implant mucositis, alternative/additional measures (e.g., air polishing with low abrasive powders) may improve efficacy versus conventional treatments at sites of peri-implantitis.
Daubert M D et al., 2019 [14]	To investigate the role of biofilm and its removal in the management of peri-implant mucositis and peri-implantitis.	Clinical trials and observational studies evaluating different approaches for biofilm removal from implant surfaces were considered.	Mechanical debridement is effective in removing biofilm in both mucositis and peri-implantitis. Among the adjunctive therapies, air-polishing with glycine, the use of local antiseptics and lasers appear to be effective in short-term disease control.
ChalaM et al., 2020 [16]	compare the effectiveness of the adjunctive use of lasers for the treatment of peri-implant mucositis or peri-implantitis compared to the conventional treatment.	Mechanical debridement + Laser compared with mechanical debridement alone.	The adjunctive use of lasers in the treatment of peri-implant inflammation is effective for up to three months; there is no strong evidence regarding the long-term benefit compared to conventional treatment.
Lin G et al., 2018 [17]	evaluate the effectiveness of laser therapy with non-surgical or surgical therapy in managing peri-implant mucositis and peri-implantitis.	CSs, CCTs and RCTs evaluating the efficacy of laser therapy in the resolution of peri-implant disease were selected.	The use of laser therapy in combination with surgical/non-surgical therapy provided minimal benefit in PD reduction, CAL gain, amount of REC improvement, and PI reduction in the treatment of peri-implant diseases.
Albaker M A et al., 2018 [18]	To investigate the effect of photodynamic therapy or laser therapy in the management of peri-implant mucositis.	Clinical trials evaluating the efficacy of photodynamic therapy and laser therapy in the management of peri-implant mucositis were selected.	This systematic review demonstrated inconclusive findings to show the effect of photodynamic therapy or laser therapy in the management of peri-implant mucositis.
Sculean A et al., 2021 [20]	evaluate the efficacy of adding aPDT as an adjunct to standard therapy in the management of peri-implant disease.	the authors reviewed clinical trials and reviews investigating the efficacy of photodynamic therapy in addition to mechanical therapy in resolving peri-implant inflammation.	Limited evidence suggests that PDT may, on a short-term basis, reduce clinical signs of inflammation in peri-implant diseases.
Liu S et al., 2020 [23]	evaluate the role of CHX in improving outcomes with non-surgical management of peri-implant mucositis and peri-implantitis.	the efficacy of chlorhexidine as an adjunct to mechanical debridement was investigated by evaluating PPD, CAL and BoP.	adjunctive CHX therapy may not improve outcomes with nonsurgical management of peri-implant mucositis and peri-implantitis.
Wychowański P et al., 2021 [26]	consider the topical action of drugs and active biological substances for the management of peri-implant diseases.	the topical efficacy of some drugs and some biological substances was evaluated, such as chlorhexidine, doxycycline, metronidazole, triclosan and hybenx.	The methods currently used do not allow to establish the efficacy of the topical administration of the substances taken into consideration in the treatment of peri-implantitis.
Zhao P et al., 2021 [29]	Determine the efficacy of chlorhexidine (CHX) as an adjunctive therapy to mechanical debridement in the treatment of peri-implant diseases.	Randomized controlled trials (RCTs) comparing mechanical debridement combined with CHX to mechanical debridement alone for patients with peri-implant disease were identified.	Based on available evidence, adding CHX to mechanical debridement, compared with mechanical debridement alone, did not significantly enhance the clinical results.
Mattar H et al., 2021 [31]	the effect of using the diode laser in the treatment of peri-implantitis was evaluated.	Studies comparing the use of laser with standard therapy in patients with peri-implantitis were analysed. studies with a minimum of 6 months follow-up and which evaluated BOP and CAL were selected.	The data from this systematic review do not support a recommendation for the use of a diode laser (810 nm) in the management of peri-implantitis, although the diode laser has a scientifically proven effect of decontaminating implant surfaces.
Atieh M A et al., 2022 [34]	assess the outcomes of using diode laser on the management of peri-implant mucositis in terms of changes in periodontal parameters.	only randomized controlled trials (RCTs) comparing the combined use of mechanical debridement and diode laser with mechanical debridement alone were selected.	In the management of peri-implant mucositis, the combined use of diode laser and mechanical debridement did not provide any additional clinical benefit over mechanical debridement alone, although the laser has been shown to have an antimicrobial effect on various implant surfaces and materials. Furthermore, diode laser irradiation on the healing abutments significantly eliminated the predominant pathogenic bacteria and accelerated wound healing without any detrimental effect on the evaluated implant material.
Lin Y et al., 2022 [35]	evaluate the safety and efficacy of different lasers and PDT compared to conventional mechanical debridement for peri-implant treatment.	We only sought RCTs evaluating the clinical efficacy of adjunctive PDT, different lasers, and CMDs. Clinical outcomes were changes in probing pocket depth (PPD), marginal bone loss (MBL), and attachment level (CAL).	the adjunctive PDT achieved a small additional benefit on PPD reduction and MBL gain compared with CMD alone and had the highest probability of being ranked first on the changes in PPD, MBL and CAL. PDT + CMD may represent an alternative method for peri-implant treatment.
Zhao Y et al., 2021 [37]	The clinical efficacy of antimicrobial photodynamic therapy (aPDT) compared with antibiotics in periodontitis and peri-implantitis was investigated.	RCTs were selected in which aPDT and Antibiotics were added to SRP for the management of peri-implantitis.	aPDT can be considered as an alternative to antibiotics in the treatment of peri-implantitis and periodontitis.
Louropoulou A et al., 2011 [39]	Evaluate the effects of different mechanical tools on smooth and rough titanium implant surfaces.	Studies investigating the alterations of the implant surfaces, both smooth and rough, after mechanical instrumentation with various metallic and non-metallic instruments were taken into consideration.	Non-metallic instruments and rubber cups appear to be the instruments of choice for smooth implant surface treatment, while for rough implant surfaces, nonmetallic instruments and air abrasives are the instruments of choice, especially if maintenance is required. the integrity of the surface.
Yen Nee W et al., 2022 [44]	This study reviewed the effects of titanium implant surfaces on different hygiene instruments.	Studies investigating the impact on the planar surfaces of different instruments used during mechanical therapy were examined.	The metallic instrument should be avoided on titanium implant surfaces. A non-metallic instrument such as a plastic curette, rubber cups, and novel technology including diode laser, LED, and laser treatment is appropriate and can be used on smooth, machined, SLA, TPS, and RBM titanium implant surfaces for debridement.

## Data Availability

Data are available at the corresponding authors upon reasonable request.

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
