# Peer review of "Consensus Report of the Technical-Scientific Associations of Italian Dental Hygienists and the Academy of Advanced Technologies in Oral Hygiene Sciences on the Non-Surgical Treatment of Peri-Implant Disease"

_ijerph, 2023, doi:10.3390/ijerph20032268_

Round 1

Reviewer 1 Report

This consensus report surely interests many readers, especillay those target users of the guideline. The major problem about this manuscript is English usage. In INTRODUCTION and DISCUSSION, The authors need to put several relevent sentences into single paragraph and avoid one-sentence paragraph.

Other points:

Line41, Materials and Methods

Line43, What is GRADE method, Please deliberate in the context

Line55, incomplete sentence

Line72, "prevention" is irrelevant to this manuscript

All tables need to be redesigned to make them more clear

Line185, peri-implant-diease

Line186, peri-implantitis

Line212 and 227, Are these two Guidelines the same?

Table for Q2A, PPD (Probind Depht)

Table for Q5A, his locally usage???

Table for Q1C, "the use of grommets is not discouraged", Please do not use double negative description

Table for Q4C, What is "lice surfaces"?

DISCUSSION, the definitions of peri-implant condition appeared again. The authors need to decide where they should appear (introduction or discussion)

Author Response

Hello, thank you for your review of the manuscript. in the attached file we have replied to all your comments, you will find the corrections in the red text.

Reviewer 2 Report

I would like to thank the authors for their sustained effort. Peri-implant disease is a novel topic and there is much to be learned and studied. The authors have written a consensus and an article regarding this entitled “Consensus Report of the Technical-Scientific Associations of

Italian Dental Hygienists and the Academy of Advanced Technologies in Oral Hygiene Sciences on the Non-Surgical Treatment of Peri-implant Disease”.

The new categorization of periodontal and peri-implant disease has provided precise guidelines for diagnosing the different phases of these illnesses. However, there are no clear 37 indications for treating these disorders. This Consensus Report guides non-surgical treatment of peri-implant mucositis 39 and peri-implantitis. The latest scientific literature was analyzed for the consensus. Procedures: The 41 Italian technical-scientific societies (AIDI, UNID, and ATASIO) chose 15 experienced Italian dental hygienists to establish GRADE-based prescriptions for treating peri-implant mucositis, peri-implantitis, and implanting surfaces. Conclusion: non-surgical treatment alone can treat peri-implant mucositis but not peri-implantitis, according to worldwide research. Some additional medications may reduce inflammation.

Regarding the manuscript I have some suggestions:

1.    All the authors have RDA DHA acronym which is not explained

2.    There is stated eligibility criteria where you include studies, but no exclusion criteria are stated.

3.    A summary of the included studies would be appreciated so the reader would fully comprehend the results. It is very difficult in its current form to critically forming an idea if there is no summary to see how many patients, interventions and so on have led to the conclusion stated. There is data missing that resulted from the search and sustains the results.

Author Response

Hello, thank you for your review of the manuscript. in the attached file we have replied to all your comments, you will find the corrections in the blue text.

Round 2

Reviewer 2 Report

Thank you for the modifications made and for your prompt response. 

Congratulations!

PS: there is a slight spelling error in line 319:

DHA Doctor in Healt Administration